# Assessing Fisheries Policies of Bangladesh: Need for Consistency or Transformation?

**Md. Mostafa Shamsuzzaman** [1,*] **, Mohammad Mahmudul Islam** [1] **, Amany Begum** [1] **, Petra Schneider** [2] **and Mohammad Mojibul Hoque Mozumder** [3]

[1]  Department of Coastal and Marine Fisheries, Sylhet Agricultural University, Sylhet 3100, Bangladesh
[2]  Department for Water, Environment, Civil Engineering and Safety, University of Applied Sciences Magdeburg-Stendal, Breitscheidstraße 2, D-39114 Magdeburg, Germany
[3]  Fisheries and Environmental Management Group, Helsinki Institute of Sustainability Science (HELSUS), Faculty of Biological & Environmental Sciences, University of Helsinki, 00014 Helsinki, Finland
*  Correspondence: shamsuzzamanmm.cmf@sau.ac.bd

**Abstract:** With the aim to enhance production, alleviate poverty, meet animal protein demand, earn foreign currency and maintain ecological balance, the Bangladesh government has formulated the National Fisheries Policy 1998. Over the last two decades, this policy for safeguarding fisheries is still in practice but gets little attention by researchers and policy makers to assess its effectiveness. This study analyzes the fisheries policy frameworks and evaluates how policy changes affecting fisheries production with certain ecological balance. The paper describes elements in the historical process of the development of the national fisheries policy related to the issue of equal or restricted access to the fish resource. The findings suggest that changes in policy only could not offer solutions to prevent over exploitation and overcapitalization that presently exists in conventional open access fishery. In addition, key constraints underlying in between policies and in implementation of laws includes ignorance of conservation laws, overwhelmingly top-down decision-making, lack of appropriate policy goals, inadequate enforcement, outdated policy and bogus action strategy, lack of enforcement regulations against pollution, poor coordination and technical know-how of the personnel concerned. For achieving inclusive growth in the fisheries sector, the UN Sustainable Development Goals and the government stated the Vision 2021, fisheries policy reform is recommended with special emphasis on marine fisheries sub section formulation, socio-economic development of relevant communities, updating of existing governance, and strengthening institutional capacity to appropriately manage this potential sector. Moreover, the existing regulations should be amended accordingly with clearly defined reliable enforcement authority.

**Keywords:** fisheries policy; governance; sustainability; legal framework; food security; Bangladesh





## 1. Introduction

Fisheries is one of the most productive and dynamic sectors in Bangladesh [1,2]. It plays a significant role in income generation, food and nutrition, and earning foreign currency in the economy of Bangladesh [3,4]. This sector is the second largest employer in rural areas [4]. In the Bangladeshi national diet, fish and rice are complementary to each other, which made Bangladeshi people be called *"Maache-Bhate Bangali"*, meaning 'fish and rice carries out our true identity' [4–6]. The total production of Bangladesh was 4.27 million MT in 2018–2019, of which more than half (56.24%) of it contributed by aquaculture only [1]. At present, the fisheries sector contributed to 3.57% of Bangladesh GDP, with about one-fourth (25.30%) to agricultural GDP and 1.39% of total export earnings [1]. After independence in 1971, the fisheries production in Bangladesh has shown a consistency within an increasing trend. In the years 2017–2018, the fisheries production reached 4.27 million Metric Tons (MT), whereas, in 2001–2002, the total production was 1.78 million metric tons shown in [1]. Now, Bangladesh placed 3rd in inland open water production

in 2017–2018. The country also placed in 5th place in the entire world for aquaculture production. It also achieved 4th position in tilapia production throughout the world as well as 3rd position in Asia [2]. As the single most important species, Hilsa, the national fish of Bangladesh, contributed about 12% to the country's total fish production [2].

Policy is a document consisting of a set of goals and strategies [7] and actions by the government to control any system, to assist adjusting problems within the system or caused by it, or to help receive benefits from it [8]. Fisheries policy sets out the priorities for the national fishery sector—for example, maximizing profitability of the sector, fish production, or maintaining as many fishing jobs as possible, and fishery assessments determine what level of fishing can be sustained at each of those target reference points for management [9].

Over the periods, various development initiatives have been adopted by the Bangladesh government. The Sustainable Development Goals (SDGs) and the Blue Economy Agenda targeted boosting fisheries production to achieve foreign currency as well as to feed the nation's economy. Hence, Bangladesh has integrated the SDGs Agenda in its 7th Five Year Plan (FYP) (2016–2020) [10]. In addition, Bangladesh has taken different plan and strategies to enhance the country's economic growth in order to transform Bangladesh from a lower income to a middle-income country [11]. However, the development policies have focused mainly towards expansion, production and technology-based approaches [12]. Again, to abide by the commitment made to Small-scale Fisheries Policy and to protect fisheries, an improved fisheries policy is needed.

The necessity for policy analysis arises from the need to facilitate the choice of sound policy with a view to improvement [13]. In addition, review of the policy is expected to overcome the shortcomings in management and therefore might lay the basis for the reformation of a worth management mechanism [14]. The Bangladesh government produced different policies to manage fisheries sector; however, the National Fisheries Policy 1998 influences the management of the whole sector. Therefore, this review focuses on the main policy document for fisheries management in Bangladesh. The study on the national fisheries policy is addressing the gaps and constraints to be resolved to achieve the goal of the policy [14,15]. After 23 years of its formulation, there is still a lack of any amendment that might have been performed in order to adjust major gaps and constraints for the proper management of these potential resources. Therefore, it is necessary to analyze fisheries policy with the overall goal of sustaining fisheries resources for a longer perspective. Indeed, no review of this only policy for Bangladesh fisheries sector has been performed so far, particularly not under consideration of a long-term perspective. In this paper, we aim to analyze the existing fisheries policy, addressing major gaps and constraints of the policy to explain the implementation effectiveness and the conclusion of whether it supports the sustainability of fisheries or not.

## 2. Materials and Methods

The study was designed based on primary and secondary data. A short history and position of Bangladesh in regard to the fisheries policy were examined through the analysis of national fisheries documents and scientific papers. We review the whole policy documents in terms of execution. Furthermore, peer reviewed articles from different journals related to governance, documents of various forms, ministerial orders, etc. were studied, analyzed through online search, and relevant information was extracted and analysed. Data on catch status were also collected from websites of the relevant Ministries, Divisions and Departments e.g., Ministry of Fisheries and Livestock (MoFL), Department of Fisheries (DoF), Bangladesh Fisheries Development Corporation (BFDC), Fishery policies, Government gazettes, Acts and Ordinance, and marine and inland fishery resources of Bangladesh. A literature review was done to analyze the existing national policy in addition to a comparison of how it should be designed in order to effectively support the needs of the fisheries sector development in Bangladesh.

## 3. Results

### 3.1. A Brief Overview of the Fisheries Sector in Bangladesh

Bangladesh comprises one of the world's biggest deltas having been blessed with three major river systems i.e., the Ganges, the Brahmaputra and the Meghna (GBM) in addition to vast, diversified marine resource supports and plentiful fisheries resources with enormous development prospects [2,7]. The inland capture fishery resources contained a total of 3,927,142 ha water area of which River and Estuary covered 853,863 ha, Sundarbans and associated mangrove ecosystems 177,700 ha, Beel 1,14,161 ha, Kaptai Lake having an area of 68,800 ha, and the floodplain including the baor of about 2,712,618 ha [1]. In these regions, inland culture fishery is conducted at a total of 797,851 ha, comprising ponds of 371,309 ha, seasonal cultured water bodies of about 130,488, baor of 5488 ha, shrimp/prawn farms of 275,274 ha as well as pen culture and cage culture of 6.78 ha and 7 ha, respectively [1]. In addition, the Southern part of Bangladesh is bordered by the Bay of Bengal having rich biotic and abiotic resources. After a recent decision made by the International Tribunal for the Law Of the Sea (ITLOS) for resolving the dispute on the Bangladesh and Myanmar maritime boundary in 2012 and the decision of the United Nations Convention on the Law of the Sea (UNCLOS) on the India–Bangladesh maritime boundary in 2014 [16], the country owns further the sovereign right over 18,813 km$^2$ area in the 200 nautical miles of Exclusive Economic Zone (EEZ).

The total production of fisheries sector in Bangladesh was reported to be about 45.03 million MT during 2018–2019, showing an increasing tendency since 2008–2009 when the total production was estimated with 27.01 Million MT [1]. The sector provides 12% of the total employment for the population, directly or indirectly [1]. Currently, the inland aquaculture sector shares the highest production regardless of its limited water resources. The systematic analysis of the data from 2001 to 2018 shows a great decline in inland capture production from 68.89 million MT (38.68% of the total fisheries production) to 12.17 million MT (28.45% respectively). However, the marine fisheries, although they have a high fisheries potential within its vast water area, share only about 65.47 million MT, which contributed about 15.31% to total production [1], showing a small increase over the analysed period of 2001 to 2018 (Figure 1). In addition, about 0.5 million fishers are involved in coastal and marine fishing using approximately 67,699 mechanized and non-mechanized boats [17]. However, the fisheries sector of Bangladesh experiences different constraints (Table 1).

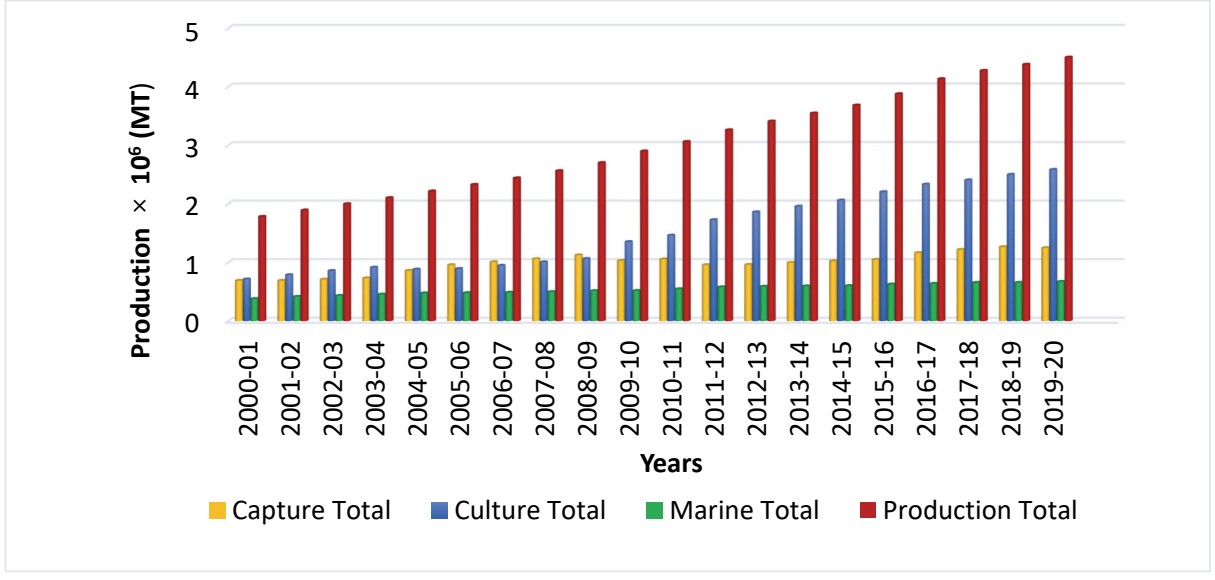

**Figure 1.** Sector wise annual fisheries production from 2000 to 2020 in Bangladesh [1].

**Table 1.** Constraints of the Bangladesh fisheries sector as identified by FAO [2].

| |
|---|
| **Catch decline:** Though this sector has major water areas, gradual decline in catch is a major concern in case of overexploitation and low population recruitment restricting the colossal flood control, irrigation and drainage structures. |
| **Food security:** Overfishing causes the food chain and food production in this sector to dwindle, resulting in food security being a major concern for present and future days. Other factors such as discarding, high grading, culling of catches, and capture of high levels of trash fish in both the coastal and offshore fisheries may have a substantial negative impact on food security in the long-run. |
| **Migratory disturbance:** Restricting migration of hilsa due to harvesting of mature and immature fish, changes to the migratory routes, damaging of spawning grounds resulted in hilsa stock decline. |
| **Increasing efforts followed by a reduced CPUE (Catch Per Unit Effort):** Artisanal fisheries are overcapitalized leading to a decrease in CPUE due to overcrowding. |
| **Problems in legislative framework:** Absence of clear policy, guidelines and strategy, ambiguity and inconsistency between policy, legislation and the management procedures and practices. |
| **Problems in governance:** Governing exploitation, development, management and conservation of fisheries resources are inconsistent as most of frameworks are outdated, less amended, have not yet come into action due to lack of coordination, and have weak enforcement, non-compliance of laws, lack of manpower etc. |
| **Lacking statistical records:** Is insufficient to manage the sustainability of resources. |
| **Pollution:** Threatened the existence of fish and their habitat. |
| **The lack of easy access to credit:** One of the problems facing rural women who work in post-larvae shrimp collection. |

### 3.2. History of Fisheries Policy Reform Efforts in Bangladesh

Fisheries resources of Bangladesh are categorized into open water and closed water bodies, which is managed under the government. Government developed policy, acts, laws, and rules to regulate the sector over time. First, to accelerate the leasing of water bodies, a committee, namely Jalmohal, was formulated to have control over it. The lease value was determined by the Ministry of Land (MoL) while the management approach was taken by the MoFL. This practice facilitated the local muscle men with a lease water body and thereby employ fishers on it. From then, the exploitation of poor fishermen was continued to secure the interests of leaseholders, resulting in a gradual decrease in fisheries production, adversely impacting socio-economic conditions of poor fishers [18]. To increase fisheries production and establish fishing rights of actual fishers, the new fisheries policy was introduced in 1986. The main objective of it was to divert maximum benefit to the fishermen. The approach was developed based on the slogan '*Jal Jar Jala Tar*', which means he who has fishing gears has access to water bodies for fishing. Along with MoFL and DoF, a fishers' association, namely, Jatio Matshajibi Samiti, volunteered in identifying the so-called genuine fishers. However, the leasing system for flowing rivers were sabotaged and fishing has been made open for all at free of cost except with mechanized boat users since 1995. The policy was formulated, however, to benefit the actual fishermen, but actually had no action at all. According to the abolition of leasing rivers, the policy was dismissed but was further revived as the National Fisheries Policy (NFP) with the initiatives of Jatio Matshajibi Samiti and DoF in 1998 for ensuring sustainable fisheries management and has been in action up to the present date.

### 3.3. New Fisheries Management Policy (NFMP, 1986): Lessons from the Failure

Prioritizing the rights of genuine fishermen in the fishery with licensing was firstly introduced in 1986 through a proposed management system of Jalmohal, namely the NFMP. The policy was focused to enhance fisheries production as well to sustain fisheries. The objectives included diverting maximum benefits to fishers and to adopt conservation initiatives to strengthen sustainability. In the policy, the genuine fishers were given a central position, intended to increase the development of the fisheries sector by limiting exploitation of the poor. The policy was to be implemented by: (i) identifying so-called "genuine fishers" and organizing them into groups, (ii) providing licenses for well-defined waters together with the gradual phasing out of the yearly lease system and (iii) the

provision of technical inputs to these groups. According to policy sentences, actual fishers are supposed to achieve licenses. While a *bona fide* fisher is he who earns his maximum income from fishing, the term *bona fide* fails to differentiate a non-fisherman from a fisher that is getting involved as a non-fisherman in the changed purview. Moreover, the NFMP was based on the slogan '*Jal Tane Je, Jola Tar*' (he owns the water body who works with a gear). However, realistically, a fishing elite has developed who are not directly involved in the fishery but earn a lot through hiring out the gear which they own, or engaging fishers as day laborers. The policy is not addressing the issues; thus, social equity will probably never be achieved. Though the policy was introduced to ensure direct access of the fisher flock, nevertheless, identifying actual fishers and compiling lists of such fisher flock remain a complex problem. In 1995, the leasing system for flowing rivers was abolished, and fishing was declared open to all free of cost except to those who catch fish by using mechanized boats. The policy was established for the benefit of the poor fishers, but, as there was no control, fishing pressure increased disproportionate to the resource availability in the rivers, thus threatening the fish stock. Additionally, influential persons and Mastans (local musclemen) have been reported as controlling the rights to the river Jalmohal in some areas and to have harassed and exploited fishers. (The secondary stakeholders, *Jatio Matshyajibi Samity* and *Khudra Matshayajibi Samiti*, expressed this view). Following the abolition of the leasing of flowing rivers, the NFMP became inactive and finally was closed.

*3.4. Major Features of the National Fisheries Policy 1998*

The policy includes all water bodies broadly designed under four ranges like Policy for procurement, preservation and management of fisheries resources of the open water bodies; Policy for fish culture and management in closed freshwater bodies; Policy for culture of shrimps in coastal regions; and Policy for exploitation, conservation and management of marine fisheries resources. Finally, in 1998, the Bangladesh National Fisheries Policy was established in order to meet the policy guidelines lacking for fisheries resources, also because this sector could not reach the formerly projected development due to various natural and manmade causes (Table 2).

**Table 2.** Major features of the National Fisheries Policy 1998.

| Range of Policy | Major Aspects Covered | Description |
|---|---|---|
| Inland Open Water (section 6) | Prohibition | • Harmful waste discharge into water is a punishable crime (section 6.7)<br>• Banned monofilament gill net (section 6.8) |
| | Regulation and Prescriptions | • Ban harmful chemicals and insecticides (section 6.7)<br>• Engage fishermen and local government in the implementation of acts (section 6.9)<br>• Banned small size hilsa catch (section 6.6) |
| | Responsibilities | • Department of Fisheries (section 6.2) |
| Inland Closed Water/Aquaculture (section 7) | Regulation and Prescriptions | • Barren ponds to be brought under culture (section 7.6)<br>• Develop soil maps to prescribe lime and manure requirement (section 7.7) |
| | Responsibilities | • Training will be provided by the DoF and NGOs (section 7.4) |
| Coastal Shrimp and Aquaculture (section 8) | Prohibition | • Ban shrimp culture along the mangrove forest (section 8.4)<br>• Ban shrimp harvesting during the breeding season (section 8.9) |

**Table 2.** *Cont.*

| Range of Policy | Major Aspects Covered | Description |
|---|---|---|
| | Regulation and Prescriptions | • To ensure shrimp production maintaining ecological balance, consultation will be made with MoFL (section 8.20) |
| Marine Fisheries (section 9) | Prohibition | • Prohibit trawling along coastal waters (section 9.1)<br>• Ban shrimp harvesting along breeding grounds and migratory routes (section 9.2)<br>• Restrict expansion of trawler along fleet (section 9.2) |
| Fisheries Quality Control (section 10) | Prohibition | Restrict fish or shrimp transportation by open vehicles (section 10.2) |
| | Regulation and Prescriptions | • Hygienic conditions to be followed in all fish markets (section 10.2)<br>• Laws to be formulated to prevent untreated industrial waste disposal (section 10.10) |
| | Responsibilities | • Fish landing centers should have quality control facilities with prior permission from DoF (section 10.1)<br>• Fisheries Quality Control Officers will be empowered for supervision and identification of low graded fish and enforcing law (section 10.2)<br>• The government will facilitate shrimp–fish exporters and private organization's (section 10.4)<br>• The MoFL will control all development, conservation, distribution and management aspects of fisheries' resources (section 10.9) |
| Trade related (section 11) | Prohibition | • Import and sales tax will be reduced for fish harvesting equipment, export-oriented activities would be tax exempt (section 11.3)<br>• Import of exotic species and fry will be restricted without prior permission of the government (section 11.10) |
| | Regulation and Prescriptions | • Emphasis will be given to establish new fisheries related industries (section 11.2)<br>• Quality Control System of the DoF will be strengthened (section 11.4) |
| | Responsibilities | • The DoF will be the authority to issue, cancel, renew licenses (section 11.1) |
| Implementation related and others (section 12) | Regulation and Prescriptions | • A National Fisheries Council will be set up to execute the policy (section 12.2)<br>• New Laws will be formulated, existing laws will be rectified and fish acts will be appropriately used (section 12.3) |
| | Responsibilities | • The government will be the main actor for the development and execution of the programs' research, extension, training, consultancy, services and supervision (section 12) |

3.4.1. Open Water Fisheries

The policy designated guidelines regarding sound management of inland open water. To increase production while ensuring the securing the socio-economic conditions of poor farmers as well to earn foreign currency, this policy recommended conserving open water fish habitat as well as promoting culture practices along wetlands. According to article

(section 6.2) of the policy, the declaration of fish sanctuaries covering parts of significant habitats or entire wherever necessary will be undertaken to increase production and to conserve biodiversity. Breeding grounds of freshwater giant prawn, and the national hilsa fish will be conserved with a comprehensive implementation of the Fish Conservation Acts (section 6.5 & section 6.6). Moreover, conservation of threatened and environmentally endangered fishes was highly emphasized (section 6.15). The policy recommends taking appropriate care during implementation of development projects including flood control, irrigation and drainage (FCD/1) projects (section 6.1). The policy also emphasized fish culture initiation on a priority basis into haors, baors and beels after renovation, water logged areas due to flood control and irrigation projects, with government khas water bodies ensuring appropriate use of all water bodies (section 6.4, sections 6.10–6.13). Indeed, this policy recommended a new integrated model for fish/shrimp cum HYV rice culture into haors, beels and other flood affected areas (section 6.3).

### 3.4.2. Aquaculture

Nowadays, aquaculture is a growing sector in Bangladesh [19]. In 1995–1996, aquaculture production was 3.13 Lakh MT, but over time, the National Fisheries Policy has mostly facilitated this sector, and now Bangladesh is in 5th position for aquaculture production all over the world [2]. Now, closed water fisheries are the mostly recognized sector in Bangladesh contributing to about 2.4 million MT (56.24%) in 2017–2018 [1]. For the development of the aquaculture sector, this policy foresees to expand technologies. In order to do so, fish farm demonstration projects established by the government in coordination with public sectors were emphasized. This policy also empowered women's participation in fish culture along closed water bodies. Accordingly, fishers should be provided with training on culture technique and fertilizer use with emphasizing the female participants to involve themselves in culture (sections 7.2, 7.3 and 7.12).

### 3.4.3. Coastal Shrimp and Aquaculture

Shrimp aquaculture in Bangladesh expanded from the southeastern to southwestern coastal regions [19] in the 1970s, when shrimp aquaculture was started in ghers [20,21]. After adaptation of the National Fisheries Policy in 1998, the shrimp farming was supported to be expanded along coastal areas through improved technologies. This policy emphasized the zoning of coastal areas for shrimp culture, adopting integrated shrimp cum rice culture, improved-extensive culture, development of appropriate technologies, quality maintenance during feed production as well as post-harvest handling of shrimp produced, with providing facilities through expansion of Quality Control Laboratories, and establishment of commercial shrimp hatcheries through private entrepreneurship on a priority basis (sections 8.2, 8.4–8.8, 8.10–8.13, 8.17–8.23). This policy prescribes to declare for a ban period for brood shrimp conservation aiming to improve shrimp production, suggested for quality control measures, supporting appropriate technological development for production to be increased, post-harvest technology as well hygienic maintenance in shrimp industries, and the export market for shrimp and shrimp products (section 8.3, section 8.9, and sections 8.14–8.16).

### 3.4.4. Marine Fisheries Sector

The policy was executed for a proper management and ensures greater production in line with the resource sustainability. The NFP 1998 emphasized the previous survey that was done by national and international cooperation to identify the resource specifics to then be transferred to local and industrial fishers. The NFP 1998 is vital for coastal fisheries conservation, management and development. The fishermen community is proposed to be involved in information gathering. Shrimp and fish harvest up to 40 fathom depth were stated to ban conducting research under projects accordingly. The policy gives the rights for small-scale fisheries to harvest fish from the coastal region, but the government fails to draw Bangladesh coastal fisheries boundaries.

### 3.4.5. Legal Issues

Section 4 of the NFP 1998 represents the legal responsibility. In section 4.1, it declared all autonomous organizations and multinational institutions, NGOs, voluntary organizations and individuals who were employed within the boundary of Bangladesh. For the development of the fisheries sector, all sectors directly and indirectly involved in fisheries from harvesting, preservation to export–import industries will fall under this policy. In section 4.2, all water bodies are announced as belonging to the policy for fisheries production, management, development and conservation accordingly. Though the DoF is responsible for the enhancement of the fisheries sector, legally the owner of the water bodies are other government agencies.

### 3.5. Laws and Policies for Supporting Fisheries Management in Bangladesh

The Bangladesh government formulated several acts, rules, laws, regulations and policies over time (Table 3) to manage and regulate fisheries resources wherein only a few are implemented to some extent to manage and regulate fisheries resources as well as to protect fish species [15]. A total of 27 acts, laws and regulations were identified in this study, governing fisheries resources throughout the country, whereas 17 have been executed to strengthen the shrimp sector to earn foreign currency [22]. The country's first fishery act was established in 1950. Most of the rules and laws were made in the last two decades following the adoption of the National Fisheries Policy in 1998 [19]. The inland capture fisheries are governed within 12 laws enacted in Bangladesh [15], whereas the vast marine water resources obtained only three acts and ordinances [23]. The proper implementation of these laws and policies is considered to be limited due to a lack of clear policy guidelines; lack of appropriate legislations; lack of a clear strategic direction; an inadequate existing regulatory framework; the lack or poor enforcement of regulations; political interference, the lack of community involvement in the management program, ignorance of conservation laws, exploitations by middlemen, corruption and irregularities of authorities and the absence of regular law review and updating mechanism of by-laws, rules, orders and so on [23,24]. All this leads to a weak management of inland and marine fishery resources throughout Bangladesh. Moreover, various problems were identified by the NEMP (National Environmental Management Plan) including overfishing and exploitation of limited resources, though the impact of these is less noticeable and also injudiciousness and mishandlings in leasing water bodies are reported extensively, thereby outwitting the intent of the policy [25].

**Table 3.** Overview of key legislations regulating the fisheries sector in Bangladesh [1,19,23,24].

| Title of the Laws | Aspects Covered |
| --- | --- |
| The Private Fisheries Protection Act 1889 | Protects the private fisheries (Fishponds and *Jolmahals)* by legal framework |
| The Protection and Conservation of Fish Act 1950 and Rules 1985 | Conservation of fisheries resources as a whole. Rules regulate protection and conservation of fish |
| Territorial Water and Maritime Zone Act 1974 and Rules thereunder 1977 | Marking boundary and conservation of marine fisheries |
| Marine Fisheries Ordinance and Rules 1983 | Management, conservation, and development of marine fisheries. Rules regulate the issuance of fishing licenses |
| Bangladesh Environment Conservation Act 1995 and Rules 1997 | Conservation of natural resources and ensure eco-friendly development. Rules regulate the management of ECAs |
| Embankment and Drainage Act 1952 | Protecting crops, not allowing cuts in embankments to produce shrimp |
| Bangladesh Water and Power Development Board Ordinance 1972 | Develop water management infrastructure for shrimp farming |
| Bangladesh Fisheries Development Corporation Act 1973 | Development of the fishing industry of Bangladesh |

**Table 3.** *Cont.*

| Title of the Laws | Aspects Covered |
|---|---|
| Protection and Conservation (Amendment) Ordinance1982 | Concern on definitions and technical matters |
| Fish and Fish Product (Inspection and quality control) Ordinance 1983 and Rules 1997 | Quality control fish and shrimp targeting export |
| Manual for Land Management 1990 | Allocate unused state (khas) land to the landless on a permanent or temporary basis |
| Shrimp Estate (*mohal*) Management Ordinance 1992 | Allocate suitable state (khas) land for shrimp farming |
| Shrimp farm taxation Law 1992 and Rules 1993 | Imposing higher tax on shrimp land to cover cost of polder infrastructure |
| National Fisheries Policy 1998 | Increasing production, maintaining ecological balance and fish conservation |
| Fish and Animal Food Act 2010 | Safe fish and animal feed production, processing, quality control, import, export, marketing and transportation |
| Hatchery Act 2010 and Rules 2011 | Sustainable hatchery development to ensure quality fish and shrimp seed |
| Pond Development Act 1939 | Management of barren ponds for fish culture and irrigation |
| The Shrimp Plot Lease, Renewal, Management and Development Policy 2013 | Enhance shrimp production and export, disseminating improved technology and ensuring environmental sustainability |
| National Shrimp Policy 2014 | Sustainable and ecofriendly production of shrimp and enhanced export earnings |
| Code of Conduct: For Various Segments of the Aquaculture-Based Shrimp Industry in Bangladesh | Sound and sustainable development of the industry ensuring environmentally sustainable and socially acceptable shrimp and other fisheries products |

*3.6. Major Constraints of the National Fisheries Policy 1998*

The government of Bangladesh has formulated the national fisheries policy which provides the legal framework for better managing of the fisheries sector and contributing to the nation's economy. The policy mainly focused on enhancing production of the fisheries sector. Indeed, it emphasized biological and ecological resource sustainability as well as social and economic aspects of the fisher community. Even the national fisheries policy is rich in a strength, management regime, and it is not supported by the required actions of putting it into practice. The key constraints identified in this study that may result in ineffective fisheries management are shown in Table 4.

**Table 4.** Identified constraints of the national fisheries policy 1998.

| National Fisheries Policy 1998 | Major Constraints Identified |
|---|---|
| **Gaps in Policy Sentence/Texts** | Lacking appropriate policy goals |
| | Overwhelmingly vertical top-down participation in decision-making |
| | Lack of community centered management approach |
| | Ignores conservation issues |
| | Influenced by particular species and gears |
| | Absent sentence against aquatic pollution |
| **Gaps in the Execution of Existing Policy** | Haphazard fishing and continued over-exploitation |
| | Conflicts among different sectors and ministries |
| | Inadequate enforcement of fishing regulations |
| | Outdated policy and carrying out by so-called action strategy |
| | Too few addressing of protected areas for fishing |
| | Absence of proper orientation and technical know-how of the personal concerned |

- *Indiscriminate fishing and continued over-exploitation:*

  The expansive, interconnected river systems of Bangladesh are gifted with a very rich biodiversity of many species. Presently, many species are in danger of extinction due to over exploitation and abuses [26]. Unfortunately, though the policy mentioned securing access rights of original fishers, they are underpinned by the local musclemen or political omens providing money for leasing in advance. As a result, overfishing in the Jalmohal has become a common practice. With the purpose of covering the lease cost price and obtaining maximum profit, the contracts tend to allow the harvesting of as much fish as possible without any consideration of resource sustainability. Indeed, users of the island still haphazardly utilize the natural resources of coral reefs. The destruction of habitat and over-exploitation of these resources have resulted in declining the biodiversity [26].

- *Conflicts among different sectors and ministries*

  Conflicts among different ministries are a major problem of fisheries resource management. In the case of ownership by the government, khas wetlands were identified to represent difficulties with ownership of land and water. Management and conservation of Sundarbans wetlands for aquatic biodiversity conservation face difficulties due to conflicts among the empowered authorities [27]. The wetlands and water bodies managed and conserved are headed by the FD officials and, under the department of the environment, however, they receive little attention from the responsibilities. Though the poor people are immediate gainers for violations of the laws, marine fisheries resources are illegally extracted and exploited by the influential middleman, supported by politicians and forest bureaucrats. There is a lack of intra and inter departmental coordination resulting in widespread inefficiency [24]. The FD has resisted sharing any functions with the DoF officials, local users, NGOs and other government agencies.

- *Lack of Community Centered Management Approach (CBFM)*

  The NFP 1998 calls for production-based management of open water fisheries as opposed to leasing. The Policy commits to promoting the involvement of poor and traditional fisher folk in the management and conservation of both open and closed water bodies, but it does not mention CBFM as an approach. Even the policy does not provide a strategic direction how the actual fishermen would be identified and in which way they would be involved in the management of water bodies.

- *Ignoring of conservation issues*

  The management of *Jalmohal* mainly focused on accruing revenue from fisheries production, allowing the leasing of water bodies to fishermen or local communities [15,18]. The leasing contract of wetlands in Bangladesh found the following three categories; the lease allowed for 3 years, for 6 years under a development scheme as a maximum, though this was allowed for 5–10 years in the past under other ministries development projects rather than the MoFL [3]. Because of the short-term leasing nature in most fisheries, the leaseholders have no incentives to undertake conservation measures for rehabilitation of the stock or preservation [3,24]. The system, therefore, contributes only to a small extent to rational and long-term fishery management. Moreover, the existing fisheries and marine fisheries policies were formulated decades earlier and comprehensibly these did not reflect and/or incorporate some important marine ecosystem conservations like marine protected areas (MPA), marine reserves, species protection, coastal water quality, biodiversity protection, pollution control, stock assessments-based exploitation, poverty of coastal dwellers and climate change [23].

- *Overwhelmingly Top-Down Participation in Decision-Making*

  The primary and vital policy setting in fisheries in general and marine fisheries policy formulation in particular typically remains with the technical government staff in the fisheries ministry/department of fisheries and a single research institute staff at the national level with occasional participation by a few university faculties. Non-sectoral

political government staff, NGOs, academicians and other stakeholders are less represented at the national level. The technical fisheries staff at district and local levels are the next most influential group.

- *Influenced by Particular Species and Gears*

In the existing fisheries policies, even those not mentioned specifically in the documents, but in marine fisheries related policies, there is an influence by the interest of certain kinds of fisheries (most prominently hilsa and black tiger shrimp). For instance, Bangladesh with the important hilsa and shrimp fisheries are thus bound by certain resolutions on operations of fishing vessels, license etc., and even seasonal protection through the declaration of fishing bans [23]. Certain important fisheries may have an influence on national policy development, either through the need to conform to certain conditions or as a result of raised awareness developed and consequently representative participation by people associated from these groups.

- *Lacking Appropriate Policy Goals*

The fisheries policies together with other associated policies appear to be mainly focused on the outcome [15]. However, marine environmental policy targets should rather focus on 'outputs' than 'outcomes' because an outcome-based policy has less chance of sustainability potential. It does not comply with scientific approaches to look at the area of a certain habitat to be demarcated as protected or the number of management plans prepared rather than the outcomes in terms of increase in biodiversity or improvement in water quality; even protected areas and management plans are tools to achieve those goals [12,23]. However, firstly, the goals to be achieved are to be defined, supported by a comprehensive set of tools. All polices, management plans and interventions should be thoughtfully planned based on those definite policy target/s and rationally executed to acquire the desired results.

- *Inadequate enforcement of fishing regulations*

When fisheries regulations exist, they are not always implemented or enforced [23]. Since the DoF has no exclusive control over the fisheries, a meaningful enforcement of the Fish Act is difficult as enforcing responsibility lies with the leasing agencies and other ministries of government [24]. Officials from other departments and ministries, therefore, pay little attention to the fishery resource management [23]. Lack of political will is also responsible for failures to adopt and implement the measures taken for governing and conservation of resources [24]. In many fisheries, current rules and regulations are not strong enough to limit fishing capacity to a sustainable level. This is particularly the case for the high seas, where there are hardly any international fishing regulations to prevent illegal, unregulated and unreported fishing [24].

- *Outdated Policy and implementation by so-called action strategy*

In Bangladesh, marine environmental policy planning as such does not exist and therefore the department of environment formulated the policy in a wide way with some specific environmental issues, although a number of these are now rather outdated. In most cases, it is implemented through some form of national action plan which usually performed to address sudden crises, and is often forgotten when the crisis situation is somehow overcome. In existing fisheries and marine policies, "coastal and marine policy" is scarcely specified as separate policy, and is apparently included in a wider environmental policy with some elements to be found in sectoral policies.

- *Absence of penalties against Aquatic Pollution*

In most of the policy papers, environmental pollution received scarce attention. The Bangladesh environment act banned hydrocarbon emission related air pollution, and this is remotely responsible for water pollution or marine pollution. In the Environment policy as of 1992, it is one of the subsections advocated for the prevention of aquatic and marine pollution and to preserve coastal and marine ecosystems. The Bangladesh Environment

Protection Law 1995 also outlined the need to restrict the emission of hydrocarbon related air pollution by machines and vehicles. The Environmental Pollution Control Ordinance, 1977 (Ordinance XIII of 1977) was replaced by the Bangladesh Environment Conservation Act 1995.

- *Too little attention to protected areas*

Protected areas (sanctuaries) and no-take zones are places where fishing is banned or strictly regulated. Their purpose is to provide essential safe havens where young fish can grow to maturity and reproduce before they are caught. However, in response to total inland open water and vast marine areas, there are only a few areas declared as sanctuaries or MPAs under the Fish Protection Act 1950 and Marine Fisheries Ordinance 1983 [12,26]. There is currently a lack of especially considering fish spawning grounds in inland and in the deep sea, both of which are particularly vulnerable to overfishing [12,15,27].

- *Absence of proper orientation and technical know-how of the personal concerned*

In many cases officials of the MOL including the Deputy Commissioners is unlikely to have sufficient academic background to understand the renewable resource characteristics of most fisheries and their vulnerability over time. Equally, the policy of leasing out fisheries to middlemen does not explicitly consider *Jalmohal* as renewable resources. Additionally, the forest reserves of Sundarbans, wildlife sanctuaries in the Sundarbans as well as along the Bay of Bengal are managed under the Department of Forests. Despite having legislative support, the fisheries and wildlife resources of the Bay of Bengal and Sundarbans protected areas receive little attention due to the lack of directional authority [27].

## 4. Discussion

Bangladesh fisheries shows an increasing production trend and more than half of this was contributed from aquaculture (Figure 1). Despite being blessed with inland open water areas of about 3,927,000 ha and a vast marine water area especially after winning two verdicts with Myanmar and India, these two subsectors, however, contributed 28.45 percent and 16 percent, respectively, to the total production [1]. A substantial imbalance in per area production rate was identified that has not yet been put into political focus. The government of Bangladesh has tried for a long time since its independence to manage this sector. Many policies, laws, rules and regulations were introduced, but implementation has not succeeded to achieve the goals. The government after its independence has targeted the boosting of production to alleviate poverty, the demands for animal protein and increasing the export earnings rather than developing sustainable fisheries. These revenue-oriented targets therefore led to a decline in open water fisheries as well as in the coastal and marine fisheries sector.

To regulate the capture water fisheries, the Bangladesh government introduced many governance tools including laws, rules, policies, ordinances from time to time (Table 3). This governance framework has been executed by allocating fishing rights to local fishers through periodical leasing. However, the efforts of allotting fishing rights are hindered because of political intervention, corruption and ignorance as well as conserving responsibilities by other empowered ministries and departments rather than the DoF, leading to overexploitation of resources and finally to stock decline [15]. This is the key barrier to implement the policy. Moreover, there is also a variety of socio-economic factors that affect the governance in its need-based, proper and timely implementation [28]. Undeniably, the NEMP (National Environmental Management Plan) has identified some concerning issues including overexploitation of resources and misuses in leasing rights that are claimed widely to circumvent the intent of the policy [15]. The policy failed to completely distinguish the policy issues and the institutional issues related to the fisheries sector [29]. Moreover, its implementation, originally intended to place the governing of water bodies with the bona fide fishers, has been largely interrupted by powerful local interests [15,30].

Several surveys inspected the status of coastal and marine fisheries resources between the 1970s and 1980s, but no latest and comprehensive knowledge is available on the fishing

stocks, biological, systematics and ecological characteristics of the coastal and marine fisheries of Bangladesh. The ESBN (Estuarine Set Bag Net, locally called *behundi jal*) fishery of Bangladesh is one of the country's important traditional fisheries, and a large population of small-scale fisherfolk has been reliant on it for a long time. The traditional, but less effective, the ESBN fishery has not only become vulnerable, being likely to be affected by other fisheries but may also be unfavorable to small Penaeid shrimp and some of the finfish resources that these other fisheries exploit. No law or legal instrument exists in favor of or against brackish water shrimp/fish aquaculture in the coastal areas of Bangladesh. There is also no existing zoning plan for coastal regions. Decisions regarding aquaculture are based on the accessibility of suitable land having ancillary facilities like proximity to water sources, transport services and the consent of the landowner for privately owned property. From 1998 to till now, almost 24 years, the marine fisheries sector underwent no initiatives of amendment, and there is even no separate policy and act on coastal fisheries of Bangladesh. Moreover, there is no information on the operation process of such zoning system, and it is not available in the policy. There are no guidelines on overloading conflict by depth zonation for various kinds of boats and vessels in the coastal areas of Bangladesh. Furthermore, the socio-economic and ecosystem impacts would need to be further addressed.

Article 9.3 of the policy is vital for the coastal fisheries conservation, management, and development. More than 90% capture fisheries is coming from artisanal or small scale or coastal fishing industry [1]. The policy suggested that extraordinary measures should be undertaken; however, there is still no management system working in coastal fisheries of Bangladesh. The policy gives the rights for small-scale fishers to harvest fish from the coastal region, but the problem government fails to draw Bangladesh coastal fisheries' boundaries. All types of fishing were banned in the six sanctuary areas for the particular period, but incentives are only given only to Hilsa fisher families. Non-Hilsa fisher families also felt this discrimination, since they suffered as well from the ban. The majority of small-scale fishers are full-time professional fishers, and their livelihood is entirely dependent on fishing. During the prohibition period, they could not fish, so their income is completely lost. There is a policy objective to expand the production in the marine sector against the recognition that some stocks may already be exploited at the maximum sustainable level. To implement the policy, there is a strong need to gain knowledge for potential expansion, and where changes can be made without threat to the livelihoods of dependent communities.

Inter-sectoral coordination is a clear intention of the policy, but the mechanisms and priorities are not spelled out. Under the NFP, the MOFL is stated to control all aspects of the fisheries sector. However, no reference was made to the necessary connection with several agencies and ministries that manage different features of fisheries. Even if the DoF is responsible for the enhancement of the fisheries sector, legally the owners of the water bodies are other government agencies. Access rights to Jalmohal larger than three acres are controlled by the MOL and Ministry of Youth and Sport. The Ministry of Water Resources is in charge of haor development. Only recently, a small number of selected Jalmohal have been handed over to the DoF to develop CBFM. However, decentralization is not unequivocally addressed in fisheries policy documents. The NFP proposes the setting up of a National Fisheries Council to implement the policy, to inspire more coordination and identifies that new laws need to be formulated in this regard. As such, the implementation plan is missing the prioritization of activities regarding capacity and requirements. It is unclear which stakeholders would benefit and how this would be implemented. Although the Policy specifically detects thirteen priority areas for action, twelve of these relate to enlarged production and export while only one relates to the 'socio-economic condition' of fishers [19]. This is an essential element in a sector where property rights' regimes and the potential social and economic consequences of improved management are highly significant to the poor. The NFP recognizes the need to carry out research and training in support of the main objectives, but a human resource development strategy is not considered, and

there is no clear statement of the priorities and responsibilities for research. Most of the laws are enforced by the department of environment and department of forests resulting in improper implementation due to a lack of particular knowledge and conflicts that arise among different sectors.

The Ministry of Fisheries and Livestock (MoFL) headed for the Sustainable Development Goal (SDGs) targets 14.2, 14.4, 14.5, 14.6, 14.7 and 14.b of Goal 14 and the Vision 2021 of the government. For achieving the SDGs targets, the NFP (1998) must be the main tool to guide the sustainable harvest of fisheries resources as well as for taking conservation initiatives. The NFP (1998) is weak regarding the SDGs goal fulfilling and long-term objectives in support of sustainability and the precautionary approach while goals do include 'maintaining ecological balance, conserve biodiversity'. The basic aim of this policy is towards a reappearance of Bangladesh in the global market for marine products, particularly shrimp. In this regard, another policy, namely the National Shrimp Policy 2014, has been formulated. The objectives and direction, however, facilitated the aquaculture expansion mostly without any consideration of environmental balance.

The Bangladesh fisheries have an enormous need and opportunity for development to support livelihoods and the national economy [4,12]. An up-to-date and appropriate fisheries policy is a prerequisite for the better planning to obtain sustainability in this sector to ensure food security for the present and the future generation. By the production boost from the aquaculture sector, the fisheries of Bangladesh share a remarkable contribution to the national GDP as well to the agricultural GDP. Responding to the targets of Vision 2021, achieving the projected production target of 45.52 lakh MT by 2021 is weakly supported by the current policy. To achieve these two targets and to sustain the average growth factor for long periods, appropriate adjustment of the policy regarding prioritization of marine resources is needed.

Identifying priority areas for improvement:

1. Institutional analysis and development of a comprehensive institutional framework with clear responsibilities, liabilities, rights and duties;
2. Policies and strategies pertaining to the sustainable use of renewable natural resources, poverty reduction and economic growth should be rationalized and enacted through appropriate action plans to be developed by the NRMC and FDEC;
3. Strategic Impact Assessment of the Policies, particularly Strategic Environmental Impact Assessment, Social Impact Assessment and Gender Impact Assessment;
4. Existing policies should be harmonized to ensure there are no overlaps or contradictions in relation to the Fishery sub-sector. Within this process, an action plan for the NFP should be developed;
5. Ensure that policymaking, which has tended to be rather top-down originating from central government favoring the priorities and interests of an influential elite, is more participatory and in line with the recommendations for decentralized government as outlined in the IPRSP document.

Promoted policies should be formulated in a more comprehensive way with cohesive strategies focusing on:

(i). Multidisciplinary and multi-sectoral approach integrating diverse livelihood strategy based on different activities and resources;
(ii). Changing attitude to include all stakeholders in the management process into a community based and co-management approach;
(iii). Recognition of small-scale fisheries as major sharers to the production and role in the social safety-net.

## 5. Conclusions

The existing national fisheries policy was identified to be insufficient to protect and conserve the fisheries resources of Bangladesh. This study reveals that the policy largely ignores the prosperous marine fisheries sector, and, therefore, wise development of this sector and its resources is a major challenge to contribute the goals of the government

Delta plan and the Blue Economy/Blow growth agenda. With ineffective management and improper support by laws and regulations, and the present development activities added extremities to these already depleted resources. Moreover, the policy prioritized the small-scale fisheries sector, but inappropriate strategic actions were formulated. Hence, the policy is weak with a view to achieving the SDGs and the Small-Scale Fisheries Guidelines.

Finally, it is important that, in adopting and implementing the policy, there is a commitment to ongoing and continued review of the policy's effectiveness in addressing ecological sustainability and the socio-economic needs of the affected communities and fishers. When the Department of Fisheries reviews the general fisheries policies, it will take the principles, objectives and management approach to the small-scale fisheries sector as spelt out in this policy into consideration to ensure alignment between the policies. Research and ongoing monitoring by both the government and stakeholders will play an important role in identifying weaknesses in the policy and its implementation. However, fisheries policies have failed to prevent overfishing because of inadequate management targets, regulations that miscarry to consider fisher incentives and actions that are not enough to unexpected shocks or disturbances. A reform in the existing policy is needed to emphasize the emerging marine fisheries with special consideration to the small-scale fisheries sector. In the case of developing a comprehensive legal framework, cultural, social and economic barriers of the targeted communities need to be considered. Better coordination among executive agencies by removing gaps and problems is another imperative step. Removing institutional weakness by appropriate logistic support is also another requirement. In every policy making, the concerned community should be consulted with. Ensuring good governance at each and every step is a fundamental requirement for sustainable utilization and management of coastal marine living resources for achieving far-reaching development objectives for peoples' wellbeing in the era of blue economy.

**Author Contributions:** M.M.S.: Conceptualization, original draft preparation, methodology, writing, editing and reviewing, M.M.I.: Writing, editing and reviewing, A.B.: Data analyzing, editing and reviewing, P.S.: Data curation, funding acquisition, and editing, M.M.H.M.: Reviewing and editing. All authors have read and agreed to the published version of the manuscript.

**Funding:** This research was funded by the University Grant Commission (UGC) of Bangladesh research grants through the Sylhet Agricultural University Research System (SAURES) at Sylhet Agricultural University.

**Institutional Review Board Statement:** Not applicable.

**Informed Consent Statement:** Not applicable.

**Data Availability Statement:** Not applicable.

**Acknowledgments:** The authors are grateful to the editor and anonymous reviewers who played a significant role in shaping and improving the manuscript.

**Conflicts of Interest:** The authors declare no conflict of interest.

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
