# Peer review of "Assessing Fisheries Policies of Bangladesh: Need for Consistency or Transformation?"

_water, doi:10.3390/w14213414_

Round 1
Reviewer 1 Report
Review report
Title: Assessing fisheries policies of Bangladesh: Need for constancy or transformation?
General comments
1. As the special issue is entitled Evaluating the Human Benefits from and Pressures on Marine and Coastal Environments – I do not see this manuscript as it is currently written fitting well. I suggest the authors when revising the paper bring forward the prime areas where policy improvement would also create human benefits that could be quantified.
2. The Methods section needs revising so that it provides sufficient detail to be able to reproduce the analysis. The authors need to provide the inclusion and exclusion criteria to arrive the set of documents that were actually reviewed with full text screening. Though the authors may not wish to include all the reviewed literature in the reference list, it should at least be available in Supplementary information.
3. In its current form the manuscript does not flow well and there are unusual words in the text that are distracting to the reader. Constancy … ? odd word do you mean consistency
4. Some introductory material is provided in the results – section 3.1 should not appear in the results section – this is introductory material. However Table 1 could stay in Results but needs text to introduce it and explain why it helps the reader understand the other results that are being presented.
5. It is confusing that the authors sometimes use the well known term aquaculture and then other times it is referred to as inland closed water bodies – it should be revised with the term aquaculture throughout.
6. In the policy analysis there are many examples in the manuscript of broad statements and then a list of sections in the policy e.g. Line 246 Laboratories, establishment of commercial shrimp hatcheries through private entrepreneurship on priority basis (section 8.2, 8.4-8.8, 8.10-8.13,8.17-8.23). The authors need to review each of these statements and check the statements really reflect the policy points and how they do or do not meet the desired criteria.
7. The key legislation that is provided may be better in Supplementary information as it is currently only adding more complexity
8. Key claims under lines 313-436 do not have any supporting references
The abstract sentence re findings in is in the wrong order and more importantly, the authors need to remove the contradictions in the abstract e.g. the sentence starting “the findings suggest changes in policy could not offer solutions” and then later “polity reform is recommended.
This manuscript while the ideas are fairly well formed, they are not compelling given the current structure. Similarly I suggest section 3.2 should be part of the Discussion. they are not restated in the conclusion and overall the manuscript would benefit from the assistance of an English editor. Some examples of words commonly used in the discipline that are not used in this manuscript e.g. sub section formulation I believe should be sub sector formulation (as the authors are talking about different fishing sectors. Multiple errors like this and which I’ve highlighted more than a dozen in the manuscript pdf make me suspect this text has been prepared using Grammarly or something similar. Please note there are many more that have not been highlighted.
Section 3.6 which raises important point regarding the constraints identified does not have any referencing – each part needs referencing to provide evidence. For example, but this is not a complete list refs need for sentences ending on lines 317, 320, 327, 331, 336, 338, etc.
Provide evidence for assertion that output rather than outcomes statement in 381-382 -
Specific comments
· Abstract unclear if the 1998 policy is the umbrella policy, provide date of the 2nd policy 1986
· Abstract 3rd sentence missing the word ‘is’ before the word still
· Abstract 3rd sentence badly worded, simplify by removing words “takes its outline”
· Abstract “pretended action strategy”?
· Abstract personal should be replaced with personnel
· Line 54 sentence is repeating what has been said in line 45
· Line 75 an improved policy seems to be required not an up to date one
· Lines 77, 79 excess brackets at end of sentence to be removed
· Line 82/84 providing one reference for this statement is insufficient and sentence
· Line 84/86 example of sentence that could be written simply but instead is confusing
· Methods … explain exactly how you selected the policy and other documents for review and whether reviewed by one, two or all authors. Provide list of all documents review in Supplementary information.
· Figure 1 remove the word scenario from the in figure caption – these are not scenarios – also consider moving to a stacked bar chart, and the colours for capture total and marine total are hard to distinguish
· Table 2 reformat for better readability but adding a column for type of aspect i.e. prohibition, regulation and responsibilities. Also remove dot points as they are distracting
· All legislation mentioned should be cited and referenced appropriately
· Types of literature reviewed e.g. gazettes, newspapers, magazines, etc. do not appear to mentioned in the references
· Line 135 about 5.16 lakh fishers … is this number correct? Surely it should be a whole number and next line remove the space in 67,699 – providing a number of mechanized versus non-mechanised would be helpful
· Line 248 what is the ban period for brood shrimp
· Line 303 remove full stop from the heading
· Table 4 remove the capitalisation of the major constraints, suggest for each constraint you provide one or two tangible examples to demonstrate how the constraint is impacting
· Line 314/315 not a complete sentence unless you add the word ‘is’ before gifted.
· Section 3.2 explain what are ‘local muscle men’ and in some case it is written as musclemen – needs consistency throughout.
· Similarly ‘political omen’ , ‘jalmohal’, ‘khas’ are terms I am unfamiliar with. Perhaps a glossary for the unusual terms in a box would be helpful.
· Check all acronyms are provided in full at first mention e.g. CBFM, NGO, ESBN, etc
· Line 443/444 sentence does not make sense
· Replace the word percent throughout with %
· Line 455/458Provide references to support claim
· Introduce the NEMP earlier in the manuscript – seems to be a key plan that the policy should be contributing to achieving
· Consider bringing forward issue of no zoning plans raised in line 469/488 into the introduction and abstract

Author Response
Comment |
Response |
As the special issue is entitled Evaluating the Human Benefits from and Pressures on Marine and Coastal Environments– I do not see this manuscript as it is currently written fitting well. I suggest the authors when revising the paper bring forward the prime areas where policy improvement would also create human benefits that could be quantified. |
This manuscript is centered on analysis of Bangladesh’s main fisheries policy. One of the overarching aims of the policy is reaping most benefits from fisheries for poverty alleviation, ensuring food security and overall socio-economic development. Thus, we believe the manuscript falls within the scope of the special issue. We also thank the reviewer’s suggestion for revision, which we gladly did. |
The Methods section needs revising so that it provides sufficient detail to be able to reproduce the analysis. The authors need to provide the inclusion and exclusion criteria to arrive the set of documents that were actually reviewed with full text screening. Though the authors may not wish to include all the reviewed literature in the reference list, it should at least be available in Supplementary information. |
Thank you for your valuable suggestion. We revised and rewrote some sentences to clarify the methodology. |
In its current form the manuscript does not flow well and there are unusual words in the text that are distracting to the reader. Constancy …? odd word do you mean consistency |
Thank you. We went through the manuscript and rewrote various sentences to give it a better flow. Yes. The word is ‘consistency’. We corrected the typo. |
Some introductory material is provided in the results – section 3.1 should not appear in the results section – this is introductory material. However Table 1 could stay in Results but needs text to introduce it and explain why it helps the reader understand the other results that are being presented. |
The section 3.1. provides an overview of the Bangladesh fisheries sector. Thus, we believe the table 1 and figure 1 would be appropriate in this section to provide the potential and challenges of the fisheries sector. A combination of figure 1 and table 1 could serve this purpose. |
It is confusing that the authors sometimes use the well-known term aquaculture and then other times it is referred to as inland closed water bodies – it should be revised with the term aquaculture throughout. |
We now used aquaculture instead inland closed water body throughout the manuscript. |
In the policy analysis there are many examples in the manuscript of broad statements and then a list of sections in the policy e.g. Line 246 Laboratories, establishment of commercial shrimp hatcheries through private entrepreneurship on priority basis (section 8.2, 8.4-8.8, 8.10-8.13, and 8.17-8.23). The authors need to review each of these statements and check the statements really reflect the policy points and how they do or do not meet the desired criteria. |
We would like thank the reviewer for such valuable suggestion. We narrate the key point where the policy focused and where emphasized. We tried our level best to justify it in the challenges as well as in the discussion section. |
The key legislation that is provided may be better in Supplementary information as it is currently only adding more complexity |
Legislations are the instruments for policy execution. We explained what are the tools available supporting and/or weakening the execution process to reach objectives. |
Key claims under lines 313-436 do not have any supporting references
The abstract sentence re findings in is in the wrong order and more importantly, the authors need to remove the contradictions in the abstract e.g. the sentence starting “the findings suggest changes in policy could not offer solutions” and then later “polity reform is recommended. This manuscript while the ideas are fairly well formed, they are not compelling given the current structure. Similarly I suggest section 3.2 should be part of the Discussion.
|
We inserted relevant supporting reference after the claims now. Indeed, we reviewed and rewrote the abstract section to better summarize the manuscript. As the section 3.2 mainly describes the evolution of fisheries policy and the role of different stakeholders. We feel this section better fit where now it is. Putting this section in the discussion section might look out of context.
|
Section 3.6 which raises important point regarding the constraints identified does not have any referencing – each part needs referencing to provide evidence. For example, but this is not a complete list refs need for sentences ending on lines 317, 320, 327, 331, 336, 338, etc. Provide evidence for assertion that output rather than outcomes statement in 381-382 - |
We would like to thank the reviewer. We inserted relevant references as suggested but not after each line. As several shortcomings are found from several common references, we added the common reference. |

Reviewer 2 Report
It is a well-written paper discussing the fisheries management in Bangladesh, which is meaningful and worth exploring. However, there are some shortcomings in the paper:
1. The paper needs a thorough froofreading as it contains numerous grammatical/typo errors.
2. The tables need to be realigned in a better shape.
3. As for the description for 'decision of UNCLOS (lines 123-124)', it is not accurate. It should be the decision rendered by an arbitral tribunal under Annex VII to the UNCLOS in 2014.
Author Response
Response to the reviewer:
Reviewer #2:
Comment |
Response |
The paper needs a thorough proofreading as it contains numerous grammatical/typo errors. |
We thanked the reviewer. We went through the manuscript to correct grammatical errors and typos. |
The tables need to be realigned in a better shape. |
We realigned all tables as per suggestion. |
As for the description for 'decision of UNCLOS (lines 123-124)', it is not accurate. It should be the decision rendered by an arbitral tribunal under Annex VII to the UNCLOS in 2014. |
We would like to thank the reviewer. We changed it in the manuscript accordingly. |

Round 2
Reviewer 1 Report
It is disappointing that the authors have not provided detailed references as I believe the approach they have taken means that this research is not able to be reproduced. I suggest that all the appropriate references are inserted in the appropriate places in the text as per my first review and not just one common reference. e.g. Key claims under lines 313-436
Author Response
We want to thank you for your valuable suggestion. We missed including references in the text according to your comments. We included a few new references, reviewed the previous ones to find the sources of claims, and now included them between texts. We also re-arranged the reference numbering in the text and reference section. You may pay a look at it from lines 313 to 436. For example, references no. 26 and 27 are newly inserted. Moreover, some claims were from previous references, so we included this in the text under particular claims.